# Extremely Low-Profile Monopolar Microstrip Antenna with Wide Bandwidth

**DOI:** 10.3390/s21165295

**Published:** 2021-08-05

**Authors:** Youngseok Ha, Jae-il Jung, Sunghee Lee, Seongmin Pyo

**Affiliations:** 1C4I Technology Planning Team, Korea Research Institute for Defense Technology Planning and Advancement, Jinju 52851, Korea; ace1002@dtaq.re.kr; 2Department of Electronics and Computer Engineering, Hanyang University, Seoul 04763, Korea; 3Department of Electronics Engineering, Hanyang University, Seoul 04763, Korea; 4Ground Station R&D Center, CONTEC Space, Sarl., Esch-sur-Alzette, L-4362 Luxembourg, Luxembourg; semi@contec.space; 5Department of Information and Communication Engineering, Hanbat National University, Daejeon 34158, Korea

**Keywords:** low-profile, broad bandwidth, degenerated modes, ground slot, conical beam pattern

## Abstract

In this paper, we propose a new monopolar microstrip antenna for a high-speed moving swarm sensor network. The proposed antenna shows an extremely thin substrate thickness supported with an omni-directional radiation pattern and wide operation frequency bandwidth. First, to achieve the low-profile monopolar microstrip antenna, the symmetrical center feeding network and the gap-coupled six arrayed patches which form a hexagonal microstrip radiator were utilized. The partially loaded ground-slots under the top patches were employed to improve the radiation performance and adjust the impedance bandwidth. Second, to obtain the broad bandwidth of the low-profile monopolar microstrip antenna, the degenerated non-fundamental TM_02_ modes, that is, even and odd TM_02_ modes, were carefully analyzed. To verify the feasibility of the degenerated TM_02_ mode operation, the parametric study of the proposed antenna was theoretically investigated and implemented with the optimized parameter dimensions. Finally, the fabricated antenna showed a 0.254 mm-thick substrate and demonstrates 1.5-wavelength resonant monopolar radiation with broad impedance bandwidth of 855 MHz and its factional bandwidth of 15.24% at the resonant frequency of 5.57 GHz.

## 1. Introduction

Recently, the Internet of Things (IoTs) technology has been evolving into a technology in which many mobile wireless sensor networks [1,2,3,4,5,6], especially vehicle to vehicle communication [7,8,9,10,11,12], swarm drone or unmanned aerial vehicle (UAV) communication [13,14,15,16,17] and inter-satellite communication [18,19], receive wireless sensor information. In these swarm sensor networks with high-speed mobility, the monopolar radiation and conical beam pattern of the antenna component are the key requirements due to the reliable communication quality of sensors existing at the same operating altitude. Additionally, the low-profile characteristics of the antenna are also issued to minimize the surface drag applied to the sensors moving at high speed. Other than these reasons, a microstrip antenna also has many advantages, such as a light weight, low manufacturing cost, relatively small size and ease of implementation and system integration [20,21,22]. Thus, the low-profile monopolar microstrip antenna is a very attractive end-terminal electromagnetic device for realizing a solid and stable wireless datalink environment in a swarm sensor network. For this reason, the state-of-the art technology of monopolar microstrip antennas has been proposed since the concept of the first monopolar microstrip antenna was announced in 1994 [23,24,25,26,27,28,29,30]. Basically, the conventional method to realize the omni-directional radiation in microstrip antenna is to utilize the equivalent magnetic loop current density of the radiator edge side, which is mathematically equal to the electric line current density. Representatively, it is divided into a method that utilizes a 1.5-wavelength over-mode resonance rather than a fundamental resonance [24,25,26] and a method that utilizes a metamaterial-based zeroth order resonant mode [27,28,29,30]. This method may show a disadvantage in that a single resonant mode supports only a single operation bandwidth, the thickness of the substrate is thick and the bandwidth is relatively narrow. And to overcome the narrow bandwidth issues, a design method of adding a slit and through-hole vias on the radiator [7,10,31,32,33], stacking multi-layered substrates for multiple resonance modes [9] and adding stubs in same substrate layer for degenerated mode with adjunct frequency with 90-degree phase delay [33,34,35,36,37,38]. Unfortunately, these technologies can have improved bandwidth, but the thickness of the antenna dielectric substrate is still thick, and the design method is complicated.

In this study, we have focused the theoretical analysis and implementation of the extremely thin monopolar microstrip antenna with broad operation bandwidth. The final goal of this study is to maximize the operating bandwidth of the low-profile monopolar antenna using two adjacent frequencies with the degenerated TM_02_ mode using a single resonator on a very thin single substrate. In other words, the proposed antenna can provide a wide bandwidth with a simple design and a thin thickness by using the degenerated mode of a single resonator on a single substrate. To verify this new physical phenomenon, that is, the degenerated TM_02_ modes, we designed the proposed antenna based on the verified antenna model previously presented in [31]. In addition, the parametric studies of the proposed antenna were investigated and optimized to control the degenerated modes of the proposed antenna. The following sections will show details of the proposed antenna configuration, electrical and radiation characteristics and electric field distribution for the operation principle.

## 2. Antenna Design and Analysis

### 2.1. Antenna Configuration

Figure 1 displays the configuration of the proposed antenna. As addressed in the Introduction section, the whole antenna geometry and denoted antenna design parameters are almost identical compared to [31] in our previous research result. In this study, we only deal with the degenerated TM_02_ modes in this work. The variables of the proposed antenna are kept almost equally compared with [31] and its description is shown in Table 1. As shown in Figure 1, the proposed antenna has a gray-color dotted center feeding patch, which is directly connected through a 50-Ohm commercial coaxial line and six black-color dotted radiating patches that are electromagnetically coupled with physical metal to metal gap capacitance. The feeding patch on the top side is hexagonal, and the bottom side has six isosceles trapezoidal ground slots. The isosceles trapezoid not only serves to secure the soldering part, but is also a key part for configuring the degenerated TM_02_ mode. Each side of the hexagonal feeding patch consists of six radiators at a distance as far as the coupling gap. It can be seen that each radiator is composed of three equilateral triangle ground slots.

### 2.2. Operation Principle and Field Distributions

Figure 2 shows the simulated frequency responses of the proposed antenna to variations of the substrate thickness, *t*. The *t* was varied from 0.127 mm (10 mils) to 0.787 mm (31 mils) with fixed *ε_r_* of 2.2 using commercially produced RF and microwave dielectric substrate. As can be seen from the results, when the *t* was changed, the dual resonant couplings varied, and lowered resonant mode was kept at around 5.0 GHz to 5.3 GHz. It was observed that the geometric configuration only determined the lower resonant mode. However, the substrate thickness of t may determine the upper resonant frequency with different coupling factors. In the case of the *t* = 0.254, the upper resonant modes clearly occurred at two different frequencies.

Figure 3 illustrates the simulated electric field distributions for understanding the dual resonant phenomenon, that is, the operation principle of the proposed antenna. As mentioned in the previous section, we used a monopolar microstrip antenna as a validated model in [31]. The omni-directional radiation is realized by utilizing the non-fundamental 1.5-wavelength resonant TM_02_ mode like the conventional monopolar microstrip antenna. And this hexagonal microstrip antenna as a validated mode was conducted on the implementation of one operating frequency band and the improvement of the radiation performance by reducing *Q*-factor using the mesh ground using one resonator in [31]. In other words, although the proposed antenna has the same configuration in [31], the degenerated mode due to the substrate thickness is not considered at all.

As shown in Figure 3a,b, two equivalent magnetic current densities in opposite directions are formed. By these two looped currents, which are mathematically equivalent to the electric line current density, the proposed antenna can form an omni-directional radiation pattern at two different frequencies. Thus, the proposed antenna showed only one single-layered single resonator with adjacent frequencies. Additionally, the identical modes, that is, the omni-directional radiations, were sustained at two different frequencies. Accordingly, the proposed antenna can accomplish the successful degenerated TM_02_ modes for omni-directional radiation. As shown in the edge patch radiator of Figure 3a,b, the electric field phase was opposite, and the electric field magnitude was identical. It can be seen that this phenomenon is the same as the degenerated mode of the dual-mode filter introduced in [32]. In detail, the degenerated mode in the dual-mode filter technology is the difference in electrical length of two diagonals, and the same resonance occurs at two frequencies that are close to each other. On a similar principle, in the proposed antenna, the degenerated mode generates two frequencies with the same electric field distribution as a result of the correlation between the horizontal resonator of the top gap-coupled patch and the vertical substrate thickness of the cavity between the top patch and bottom ground plane. Thus, the optimum dimensions of the proposed antenna can be expanded to a wide operating bandwidth by using an extremely thin substrate thickness without adding a multiple resonator, multi-layered substrate, etc. The detailed analysis results of the above two frequency regulations and couplings are further explained in following section.

### 2.3. Parametric Analysis

The parametric results of the proposed antenna are shown in Figure 4. A relative dielectric constant of *εr*, a feed patch size of *p*, a coupling gap size of *g*, a ground pad under the feed patch of *lf* and a width occurring from ground slots of *w* were investigated. For the parametric study, the following initial values of the dimension were chosen for clear degenerated modes: *εr* = 2.2, *p* = 10.0 mm, *g* = 0.2 mm, *lf* = 3.0 mm and *w* = 0.2 mm. 

In Figure 4a, the relative permittivity of the substrate, *εr*, changed from 2.2 to 6.15 with the fixed *t* of 0.254 mm. Overall, since the dielectric constant of the substrate forms a guided wavelength in which the wavelength in free space is inversely proportional to the square root of the dielectric constant, it can be seen that the resonant frequencies decreased from 5.5 GHz to 3.5 GHz as the dielectric constant increased. This is due to the fact that, compared to the length of the wavelength in the free space, the length of the wavelength in the guide wavelength inside the dielectric substrate was reduced. In particular, when the coupling gap of 0.2 mm which determines the effective dielectric constant computed from the air upper side of 1.0 and the substrate lower side of 2.2 are mathematically equal to the substrate thickness of 0.254 mm with a dielectric constant of 2.2, the degenerated modes were strongly coupled. Thus, it can be seen that all of the degenerated even and odd resonant frequencies were affected from the top front patch configuration for the lower even TM_02_ mode and substrate cavity resonances between the top patch and the bottom ground metal capacitance for the upper odd TM_02_ mode.

As can be seen in Figure 4b, as the length of six isosceles trapezoid radiators increased, which caused the one-centered hexagonal patch size to increase, both of the even and odd mode resonant frequencies of the proposed antenna decreased. The main reason for this result is that the physical patch size determines the electrical wavelength of the resonant frequencies for the proposed antenna. In other words, as the size of the trapezoidal radiator increases, the guided wavelength increases electrically due to the TM_02_ mode of the boundary condition determined by the trapezoidal radiator patch. However, the matching effect was different between even and odd TM_02_ modes, and the impedance matching results are shown in different amounts. Therefore, 11.0 mm-long patch length showed a lower resonant frequency of 4.8 GHz, and 9.0 mm-long patch length produced the highest resonant frequency of 6.8 GHz.

Figure 4c exhibits the frequency response when the gap of *g* between each patch was varied from 0.1 mm to 0.5 mm. In the case of *g* = 0.5 mm, due to the asymmetric cavity geometry that was shown in the cuboid, the *g* was 0.5 mm, the *t* was 0.254 mm and the degenerated modes merged into a single dominant frequency. However, when the *g* and the *t*, which was also affected by the effective dielectric constant, were almost identical, the identifiable even TM_02_ and odd TM_02_ modes could be separated. Additionally, as the other parameters were not changed, the couplings were kept. It can be seen that the amount of electrical coupling of the even TM_02_ mode and odd TM_02_ mode varies with the *g*, since the *g* generates coupling between top radiating patches and also controls the coupling cavity between the top patch and bottom ground. Therefore, if the operating frequencies of the degenerated TM_02_ modes are determined through the proposed antenna radiator and feeding patch, the *g* should be maximized to obtain broad operating bandwidth. 

Figure 4d displays the results of analyzing the effect of the feeding space for the coaxial connection on the bottom ground plane. As can be inferred from the results, it can be seen that *lf*, the size of the feeding pad, caused the even TM_02_ mode to decrease from 5.5 GHz to 5.05 GHz rather than the odd TM_02_ mode of 5.9 GHz. In addition, the smaller size of *lf* yielded weak odd TM_02_ mode coupling without changing the frequency changing. Therefore, the *l_F_* can adjust the maximum impedance bandwidth to expand the 10-dB impedance bandwidth based on lowering the lower even TM_02_ mode frequency and maintaining the upper odd TM_02_ mode frequency. 

From the parametric study, it can be seen that the degenerated TM_02_ modes with adjacent frequencies can be optimized from the meshed structure and the thickness of the proposed antenna. Unlike the conventional monopolar microstrip antenna with broad operation bandwidth, the proposed antenna can be designed by optimizing the degenerated TM_02_ mode with a simple, single-layered substrate and single resonator structure for the extremely low-profile omni-directional antenna.

## 3. Implementation and Measurement Results

The proposed antenna, denoted as “MK4”, was fabricated as shown in Figure 5. Using a finite element method-based full-wave electromagnetic simulator of Ansys High Frequency Structure Simulator (HFSS), the proposed antenna was finalized with physical dimensions, as summarized in Table 1. The 0.254 mm-thick substrate of Rogers RT/duroid 5880 with a dielectric constant of 2.2 and loss tangent of 0.0009 was used, and the substrate diameter, *D*, was set as 70 mm. 

All of the dimensions of the proposed antenna were optimized to maximize the operating bandwidth with the thin substrate and to minimize the fabrication errors that occur in the antenna manufacturing process. In addition, the parameters of the antenna prototype ensure process reliability. After the etching process of the antenna prototype, a commercial semi-rigid standard coaxial cable was installed with a cable length of 3 mm, excluding the SMA female connector part.

Figure 6 shows photos of the measurement setup environment of the antenna prototype for the reflection coefficient test with the E5071C Vector Network Analyzer (Keysight Technologies, Winnersh, UK) and for the three-dimensional far-field radiation pattern test.

The radiation patterns were measured in a well-defined and calibrated electromagnetic anechoic chamber with a width of 5.5 m, length of 5.5 m and height of 5 m. The anechoic chamber guarantees the capability to directly measure far-field conditions from 800 MHz to 8 GHz. 

Figure 7 illustrates the reflection coefficients and realized antenna gains of the proposed antenna. For the simulation, the 10-dB impedance bandwidth was observed at 1016 MHz from 4.910 GHz to 5.926 GHz with respect to approximately 18.75% at the center frequency of 5.418 GHz. Additionally, the measurements were taken according to the 10 dB impedance bandwidths of 855 MHz from 5.184 GHz to 6.039 GHz with respect to approximately 15.24% at the center frequency of 5.612 GHz.

The difference between the simulation and the measurement is the result of the inexact alignment between the top patch plane and bottom mesh ground surfaces in the process. In addition, errors occurred as the flatness of the extremely thin substrate was not maintained in a flat state where an isolated support was not used. In other words, in the reflection coefficient measurement experiment, it is judged that the extremely thin substrate is warped due to the surrounding air condition, which is different from the simulation result. The analysis result represented by the red dotted line considering the process error and the warpage of the substrate shows the same result as the measurement result. Therefore, it is judged that our analysis was very accurate. However, the measurement results of the fabricated antenna agreed well with the simulations of the proposed antenna for validating the degenerated TM_02_ modes. 

The realized antenna gains of the proposed antenna are shown in Figure 7b. The realized antenna peak gains were measured from 5.20 GHz to 5.84 GHz with 6410 MHz-stepped test frequencies. The realized antenna gains of 6.71 dBi were maximally observed at 5.57 GHz in 1.5-wavelength resonant TM_02_ mode. For the simulation, the realized antenna gain was 5.21 dBi at 5.81 GHz. Due to the characteristics of the wideband operating frequency, the observed realized antenna gains were 3.28 dBi at 5.20 GHz and 6.46 dBi at 5.84 GHz. As can be seen, the stable realized antenna gains were produced in a whole operating frequency band. As expected, the measurement antenna gains are almost in agreement with the simulations. It is analyzed that the reason that the measured results was higher than those of the simulation at approximately 5.6 GHz was reflected by the additionally radiated electromagnetic radiation from the 12 mm-long coaxial cable and the measuring cable connected to the proposed antenna. The deviation between the simulation and the measurement result is considered to be a reasonable value within the calibration error range occurring in the anechoic chamber.

Figure 8 exhibits the far-field radiation patterns of the proposed antenna. The radiation patterns were measured at the test frequencies of 5.54 GHz supported by the maximum antenna realized gains, which showed that the patterns were omni-directional, that is, monopolar radiations with co-polarization of the vertical linear polarization. As expected, the proposed antenna was identical to the monopole antenna mounted vertically with the radiation pattern and the polarization, as shown in Figure 8a. The measured peak gains were observed at θ = 40 degree, as shown in Figure 8b. At the test frequency of 5.54 GHz, the directivity of the proposed antenna was different from that of an ideal monopole, that is, an electrically small antenna due to the radiator of 1.5 wavelength. At θ = 40 degree, as shown in Figure 8b, the horizontal polarization (*Eφ*), that is, the cross-polarization of the proposed antenna, was observed to be −30.45 dB at the normalized radiation pattern.

The proposed antenna successfully demonstrated omni-directional radiation in the simulation results, and the hypothesis of the degenerated TM_02_ mode of the proposed antenna holds. The key performances of the proposed antenna and the recent monopolar microstrip antennas are summarized in Table 2. This study reports a minimum thickness of the substrate over 10 times higher than the previous results in Table 2.

## 4. Conclusions

An extremely low-profile monopolar microstrip antenna with broad operating bandwidth was discussed and demonstrated based on a new design approach, that is, controlling the unique degenerated TM_02_ modes from a simple substrate, a single-layered substrate and only a single resonator structure. The degenerated TM_02_ modes of the proposed antenna can be easily optimized by the periodically arrayed patch resonator with a meshed ground structure and the cavity resonance from the thickness of substrate between the top patches and bottom meshed ground structure. Due to the broad bandwidth of the proposed antenna, the numbers of swarm senor network performances can be expanded. In addition, the extremely thin substrate thickness of the proposed antenna that can even be bent flexibly may reduce the surface drag dramatically for the high-speed moving sensor nodes, such as ground vehicles, UAV swarm and LEO satellite constellations. Finally, the proposed antenna may be further studied in multiple polarization operations for the military and civil wireless communication system as an RF sensor application.

## Figures and Tables

**Figure 1 sensors-21-05295-f001:**
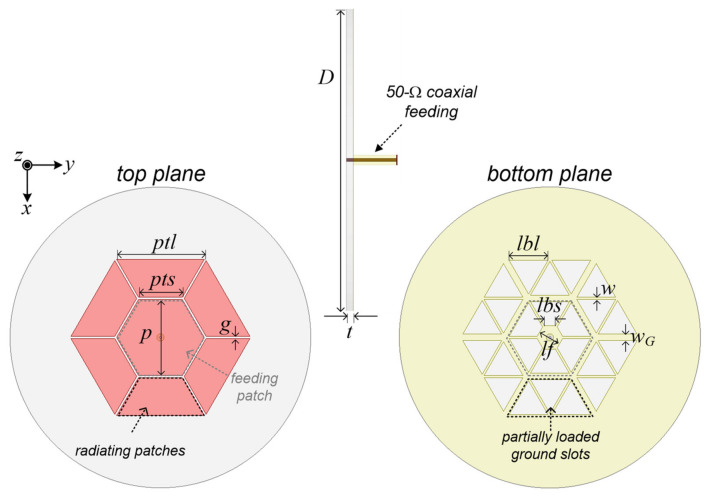
Configuration of the proposed antenna.

**Figure 2 sensors-21-05295-f002:**
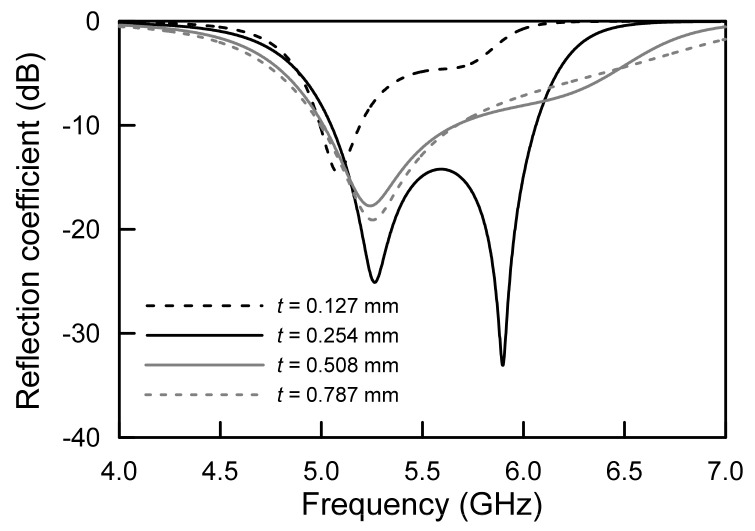
Simulated frequency responses of the proposed antenna for substrate thickness variation.

**Figure 3 sensors-21-05295-f003:**
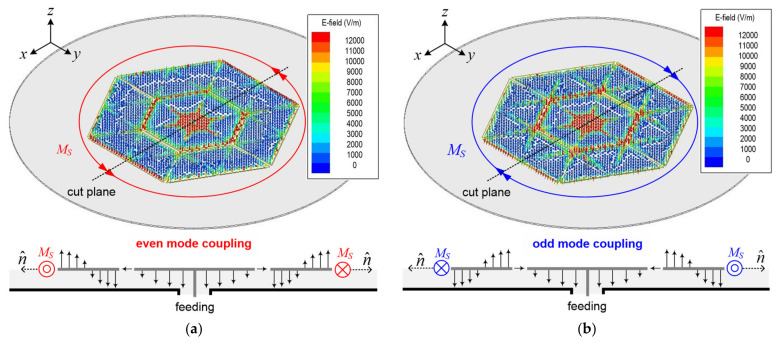
Design concept with simulated electric-field distributions at TM_02_ modes and their electric-field vector magnitudes in cut plane: (**a**) even TM_02_ mode of lower resonant frequency of 5.264 GHz and (**b**) odd TM_02_ mode of higher resonant frequency of 5.897 GHz.

**Figure 4 sensors-21-05295-f004:**
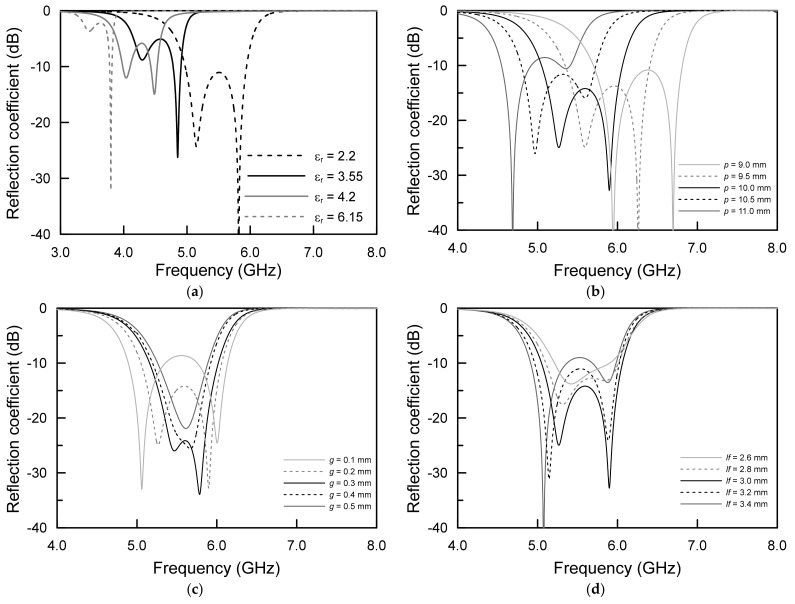
Simulated results of the proposed antenna: (**a**) relative dielectric constant, (**b**) feed size, (**c**) gap size and (**d**) ground soldering pad.

**Figure 5 sensors-21-05295-f005:**
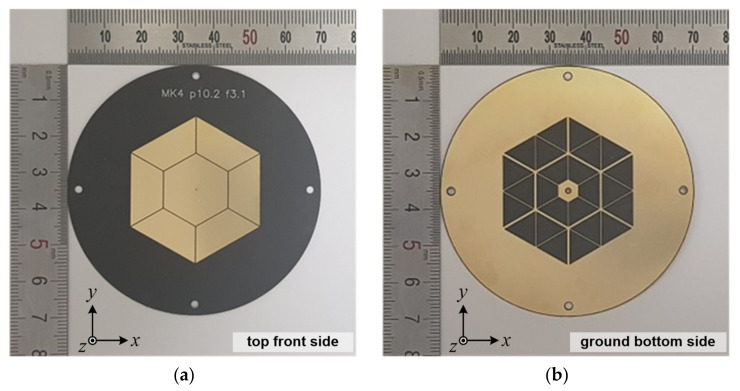
Photos of the implemented antenna: (**a**) top front side and (**b**) ground bottom side.

**Figure 6 sensors-21-05295-f006:**
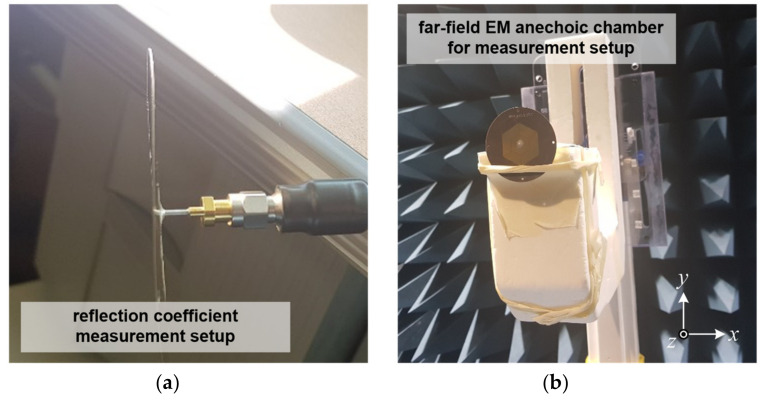
Photos of the measurement setup for the implemented antenna: (**a**) side view for reflection coefficient test and (**b**) front view for three-dimensional far-field radiation pattern test in anechoic chamber.

**Figure 7 sensors-21-05295-f007:**
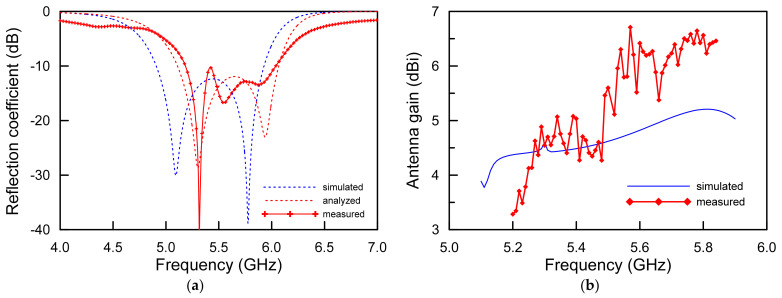
Simulated and measured results of the proposed antenna: (**a**) reflection coefficients and (**b**) realized antenna gains.

**Figure 8 sensors-21-05295-f008:**
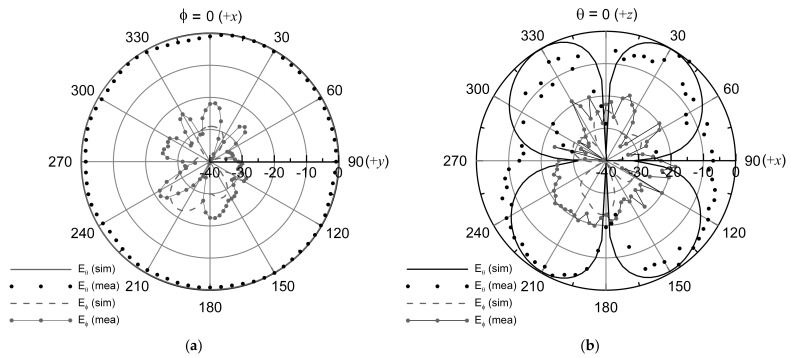
Far-field radiation patterns of the proposed antenna at the resonant frequency of 5.54 GHz: (**a**) *H*-plane (*xy*-plane) and (**b**) *E*-plane (*zx*-plane).

**Table 1 sensors-21-05295-t001:** Optimized all of dimensions of the proposed antenna parameters.

Variables	Description	Dimension (mm)
*p*	Length of the feeding hexagonal patch	17.7
*pts*	Upper length of the radiating trapezoidal patch	10.2
*ptl*	Lower length of the radiating trapezoidal patch	20.4
*g*	Coupling gap size between feed and radiator	0.2
*w*	GND wire width occurred from inner radiator	0.2
*w_G_*	GND wire width occurred from between radiator	0.6
*lf*	Soldering pad length of the feed in the GND	9.7
*lbs*	GND slot length of the upper side of trapezoid	5.4
*lbl*	GND slot length of the equilateral triangle	2.9
*D*	Diameter of the substrate	70
*t*	Thickness of the substrate	0.254

**Table 2 sensors-21-05295-t002:** Performance comparison of state-of-the-art monopolar microstrip antenna.

Ref., Work Year	Ground Size, mm^2^ (/λ_min_ ^1^)	Height, mm (/λ_min_ ^1^)	BW(%)	Antenna Type
[30], 2009	35 × 35	0.76	0.7%	Zeroth-Order Resonant
	(0.502 × 0.502)	(0.011)		
[7], 2016	π × 32 × 32	3	32.2%	Monopolar Microstrip
	π × (0.514 × 0.514)	(0.048)		
[33], 2016	π × 30.8 × 30.8	6.3	15.4%	Monopolar Microstrip
	π × (0.572 × 0.572)	(0.117)		
[10], 2019	π × 80 × 80	10	12.7%	Monopolar Microstrip
	π × (0.987 × 0.987)	(0.123)		
[34], 2017	π × 130 × 130	3.18	2.3%	Tripolarized Monopolar Microstrip
	π × (2.5 × 2.5)	(0.08)		
[35], 2019	π × 113 × 113	10.508	14.0%	Tripolarized Monopolar Microstrip
	π × (0.859 × 0.859)	(0.08)		
[36], 2019	π × 120 × 120	9.2	24.45%	Tripolarized Monopolar Microstrip
	π × (0.876 × 0.876)	(0.07)		
[31], 2019	π × 35 × 35	1.6	7.78%	Monopolar Microstrip
	π × (0.651 × 0.651)	(0.03)		
This Work	π × 35(0.651) × 35(0.651)	0.254	15.24%	Monopolar Microstrip
		(0.0044)		

^1^ λ_min_ is the free-space wavelength at the lowest operating frequency of the overlapped bandwidth.

## Data Availability

Not applicable.

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
