# Peer review of "Extremely Low-Profile Monopolar Microstrip Antenna with Wide Bandwidth"

_sensors, 2021, doi:10.3390/s21165295_

Round 1
Reviewer 1 Report
In this paper, the authors proposed an antenna that can operate at 5.57 GHz, and also can work as a sensor. An interesting design will lead to a low profile. The authors simulated and measured the results which show that both of them agree with each other. The given antenna is interesting, and also there are some important points that have to pay attention to.
- The proposed work could be consideredfor publication, but as the language /grammar of the paper is not good, It needs to be improved to increase the credibility of the proposed work.
- As mentioned in this paper, the authors only focused on the performance of the antenna, and nothing to do with the Mobile Swarm Wireless Sensor Network. Thus, this part should be deleted from the title.
- As shown in the manuscript, the omni-directional radiations can be realized this antenna. The authors should give a more detailed explanation.
- Obliviously, the special design of this antenna can be considered as introducing a frequency selection surface (FSS). The FSS is a It is a general technology, which has been reported in many papers. For instance, https://doi.org/10.1080/09205071.2021.1934571, https://doi.org/10.1002/mmce.21997. The authors can consider this to compare the differences between the those works.
- As we can see in Fig.7(b), the measured gain of this paper is larger than that of simulated results. Especially, at the point of 5.6 GHz. Authors should give corresponding physical explanations for this phenomenon.
Author Response
We appreciate helpful comments of our manuscript. We have carefully revised the manuscript based on reviewer’s helpful comments and we look forward to your positive response.
To improve readability of the manuscript, we have used official English editing service and all corrections are marked in red. The last page of the author’s response shows the official certification.

Reviewer 2 Report
A monopolar microstrip antenna with wide bandwidth for mobile swarm wireless sensor network was presented. I have the following comments for the authors:
-The writing must be improved, many sentences are missing verbs and do not make sense.
-The presentation must be improved. I had trouble to understand the design goals of this study. Please state clearly what are the design goals of this study other than monopolar radiation.
-Please state clearly the novelty of this study. Ref [31] is very similar. This study looks like a repurposed [31].
-Please state the software package used for the simulations.
Author Response

(The authors gave the same response as above.)

Reviewer 3 Report
- In the abstract, the authors state that the proposed antenna provides a "broad impedance bandwidth of 855 MHz with respect to 15.24% at the resonant frequency of 5.57 GHz". That percentage (15.24%) is usually called fractional bandwidth; moreover, "with respect" is not properly used in that context.
- The introduction is too brief. The state-of-the art should detail the contribution of each reference, rather than citing them in a group. The novelty of the work should be better highlighted by relating it to the state-of-the-art.
- Although quite acceptable in terms of gain, the results regarding the reflection coefficient do not show a good agreement between measurement and simulation. I suggest a new set of measurements on another antenna, subject to a more careful fabrication process and measuring conditions.
- The English language needs to be thoroughly revised:
- Many phrases are hard to understand
- The predicate is missing in some phrases (e.g., lines 32-34, 49-51) or does not agree with the subject (e.g., line 38).
- The use of the article "the" is not always well-suited to the context (e.g., line 46)
Author Response

(The authors gave the same response as above.)

Round 2
Reviewer 1 Report
No more comments, and I think this paper can be published.
Reviewer 2 Report
The revised version can be considered for publication.
Reviewer 3 Report
Most of the remarks and suggestions were addressed by the authors.